# Histone Deacetylase 3 Governs β-Estradiol-ERα-Involved Endometrial Tumorigenesis via Inhibition of STING Transcription

**DOI:** 10.3390/cancers14194718

**Published:** 2022-09-28

**Authors:** Guofang Chen, Qiang Yan, Lin Liu, Xinyue Wen, Hongliang Zeng, Shasha Yin

**Affiliations:** 1Shanghai Key Laboratory of Maternal Fetal Medicine, Clinical and Translational Research Center of Shanghai First Maternity and Infant Hospital, Tongji University School of Medicine, Shanghai 201204, China; 2Reproduction, and Clinical and Translational Research Center, Shanghai First Maternity and Infant Hospital, Tongji University School of Medicine, Shanghai 201204, China; 3Nephrology Center, Department of Nephrology, Zhejiang Provincial People’s Hospital, Affiliated People’s Hospital, Hangzhou Medical College, Hangzhou 310014, China; 4Institute of Chinese Materia Medica, Hunan Academy of Chinese Medicine, Changsha 410013, China

**Keywords:** HDAC3, STING, ERα, H3K4, RGFP966, endometrial cancer, therapy

## Abstract

**Simple Summary:**

Endometrial carcinoma is one of the most threatening gynecological malignancies to women’s health, with mortality linked to it increasing, and finding novel and effective methods to combat endometrial carcinoma is becoming a clinical challenge. The stimulator of interferon genes (STING) pathway plays a crucial role in antitumor immunity, and it is strictly regulated by many types of post-translational modifications. The aim of our study was to uncover the expression and the contribution of acetylation involved in the regulation of STING to endometrial cancer. We confirmed that STING expression was deregulated by both β-estradiol and HDAC3, and worked as an important regulator of proliferation and apoptosis. Inhibition of HDAC3 increased STING expression, thereby inhibiting tumorigenesis. Therefore, this study uncovers a novel molecular mechanism by which HDAC3 inhibits STING transcription via β-estradiol-ERα, and provides a promising therapy with a combination of HDAC and STING for combating endometrial cancer.

**Abstract:**

Purpose: The stimulator of interferon genes (STING) pathway plays a crucial role in antitumor immunity, and it is strictly regulated by many types of post-translational modifications. However, the contribution of acetylation involved in the regulation of STING to endometrial tumorigenesis remains unclear. Methods: We attempted to identify the key role of STING in endometrial carcinoma (EC) tissue and cell lines and explore its epigenetic regulation mechanism by HDACs that are critically involved in EC. We used IHC and qRT-PCR to detect the protein level and mRNA level of STING expression in endometrial carcinoma tissues, then explored the potential role of STING in tumor proliferation and apoptosis by CCK8 and flow cytometry, and identified the STING effect in the tumorigenicity by a mouse xenograft assay. We explored the possible relationship of acetylation alteration in STING regulation by ChIP analysis and Co-IP, and we knocked out *STING* in ECC1 and Ishikawa cells using CRISPR-Cas9 to further confirm the critical role of STING restoration induced by HDAC3 inhibitor RGFP-966 in the proliferation and apoptosis. Results: We found that STING expression was largely decreased and worked as an important regulator of cell proliferation and apoptosis; either activated or overexpressed STING, with both pharmacological and genetic approaches, largely blocked cell proliferation and induced apoptosis in EC. Moreover, STING expression was deregulated by both β-estradiol and HDAC3. Mechanically, we determined that HDAC3 can interact with β-estradiol-ERα and induce deacetylation of histone 3 lysine 4 at the STING promoter, thereby decreasing STING expression. Inhibition of HDAC3 increased STING expression, thereby inhibiting tumorigenesis. Conclusion: This study reveals a novel molecular mechanism by which HDAC3 inhibits STING transcription via β-estradiol-ERα and provides a promising therapy (a combination of HDAC and STING) for combating endometrial cancer.

## 1. Introduction

Endometrial carcinoma (EC) is one of the most prevalent gynecological malignancies originating from the endometrium. In recent years, the incidence and mortality of EC have increased year by year [1], with an estimated 61,380 new cases and 10,920 deaths in the United States in 2017. EC has recently become a major disease that is a threat to women’s health as its incidence and mortality have increased annually [2]. Although many breakthroughs have been found in combating cancer over the past decade, the five-year survival rate of patients with advanced and recurrent EC has not improved, and the mortality rate of EC patients is still increasing. The main cause for this phenomenon is primarily caused by the insensitivity of some middle–late and recurrent patients to existing treatments, such as surgery, adjuvant radiotherapy, chemotherapy, and hormone therapy [3]. Therefore, finding novel and effective methods to combat EC is becoming a major clinical challenge.

The stimulator of interferon genes (STING) was discovered in 2008 and can stimulate immunity through the action of cytoplasmic DNA. Briefly, cGAMP is synthesized by cytoplasm DNA through the action of cGAS. cGAMP, and, in turn, it interacts with and activates the STING protein lining the endoplasmic reticulum, thereby inducing STING transferred to perinuclear microsomes where the activated STING consequently activates its downstream factors, such as the NF-κB, TBK1, and IRF3 pathways [4,5,6]. Finally, immunity stimulators such as IFNβ and IL-6, and other pro-inflammatory cytokines can be stimulated by the activated STING downstream pathways [4,5,7].

STING has been revealed to play a crucial role in antitumor immunity, which mainly manifests in the activation of immune cells such as macrophages, NK cells [8], and CD8+T [9] cells. Moreover, activated cGAS–STING up-regulates PD-L1 [10], and the combination of STING agonist with anti-PD-1 therapy effectively improves the sensitivity of immunotherapy [11,12]. Cytoplasmic DNA generated during radiotherapy could activate the STING pathway, further promoting immune surveillance, and inhibiting tumors more effectively [13]. STING has become a promising candidate for tumor treatment, but it is down-regulated in various tumor types, such as gastric cancer, colorectal cancer, lung cancer, and colon carcinoma, resulting in weakened immunity and aggravated tumor growth [14,15]. The endometrium contains numerous immune cells, such as macrophages and NK cells, indicating that EC is a potential target for immunotherapy. However, the expression and regulation of STING in EC remains unclear.

Histone acetylation, a reversible modification process that regulates gene expression, is mediated by histone deacetylase (HDAC) and histone acetyltransferase (HAT) [16]. The regions with a higher degree of histone acetylation have a looser chromatin structure and higher transcriptional activity, whereas the regions with a lower degree of histone acetylation are often in a state of transcriptional inhibition. Acetylation is a key role in many physiological and pathological processes, especially in human cancers. Currently, FDA-approved HDAC inhibitors for treating liquid tumors include SAHA and VPA, whereas research on solid tumors is still in the preclinical phase. Studies have reported that HDAC inhibitors can effectively inhibit type I and type II EC [17], and the combination of HDAC inhibitors and chemotherapy has produced positive results [18]. Collectively, acetylation plays a crucial role in the progression of EC; however, little is known about the mechanisms underlying acetylation in cancers and whether acetylation regulates STING in EC.

Here, we attempt to identify the key role of STING in EC and to explore its epigenetic regulation mechanism by HDACs that are critically involved in EC with both pharmacological and genetic approaches. We provide compelling evidence that HDAC3, likely facilitated by β-estradiol-ERα signaling, inhibits STING expression and thus plays an essential role in endometrial tumorigenesis. Therefore, our study reveals an important regulatory pathway in epigenetic tumorigenesis, which has great clinical therapeutic implications.

## 2. Materials and Methods

**Cell Culture****.** The human endometrial carcinoma cell lines ECC1, Ishikawa, and H 1A cells in the lab were kindly supplied from the Key Laboratory of Maternal Fetal Medicine, Clinical and Translational Research Center, and were confirmed using STR profiling. HEK293T cells were used for the virus packaging of shRNA, and cells were maintained in RPMI-1640 medium (Gibco, Carlsbad, CA, USA) containing 10% FBS (Gibco, USA) and 1% penicillin/streptomycin (Gibco, USA), DMEM/F12 medium (Gibco, USA) supplemented with 10% FBS and 1% penicillin/streptomycin, and DMEM medium (Gibco, USA) supplemented with 10% FBS and 1% penicillin/streptomycin. All cells were maintained in a 5% CO_2_ incubator at 37 °C.

**Patients and tissue samples**. EC specimens (N = 22) and normal human specimens (N = 27) were collected from patients who had undergone surgery or biopsy procedures at the Department of Gynecology of Shanghai First Maternity and Infant Hospital affiliated with Tongji University School of Medicine from April 2016 to December 2017. Patients who were pregnant, had known immunosuppressive diseases, or were receiving immunosuppressive therapy, chemotherapy, radiotherapy, or other related antitumor therapies were excluded from the study. The patients’ information used is shown in Appendix A. The histology of all the specimens was confirmed by two independent pathologists. This study was approved by the Human Investigation Ethical Committee of Shanghai First Maternity and Infant Hospital with the approval number KS21289, and followed the ethical standards as described in the 1964 Declaration of Helsinki and its later amendments or comparable ethical standards. Written informed consent was obtained from each participant after detailed explanations regarding the study objectives and procedures were provided.

**Plasmid Construction and Transfection**. Human *STING* promoter (sequence −2000 to +299) reporter plasmid was constructed in the PGL3 basic vector at *STING* promoter sites. Five truncated *STING* promoters reported plasmids: STING1 containing −2000 to −1745, STING2 containing −1745 to −1345, STING3 containing −1345 to −945, STING4 containing −945 to −545, and STING5 containing −545 to +299 were also constructed in PGL3 vector. All the plasmids were confirmed by Sanger sequencing. Flag-tagged human HDAC3 expression plasmid (pcDNA3.1-HDAC3) and shRNA-HDAC3 plasmid were obtained from Dr. Wangsen Cao (Nanjing University, Nanjing, China). Flag-tagged human STING expression plasmid (pcDNA3.1-STING) was constructed in the pcDNA3.1 vector at Xho1/HindIII. All the plasmids were transfected into cells with Lipofectamine 2000 reagents (Invitrogen, Carlsbad, CA, USA) following the manufacturer’s instructions.

**Immunohistochemistry.** Human endometrial cancer sections were prepared, cut at 5 μm, and blocked using 5% goat serum for 1 h. STING antibody (1:1000, ab92605, Abcam, Cambridge, UK) and horseradish peroxidase-conducted secondary antibody (1:500, Weiao) were used following standard protocol. The images were captured using a light microscope. The scores for IHC were set as 3 (>50% of malignant cells), 2 (10% to 50% of malignant cells), 1 (<10% of malignant cells), and 0 (no staining) [19].

**Western Blotting.** Cells and human endometrial carcinoma tissues were lysed using RIPA buffer, and placed on ice to extract protein. Equal amounts of proteins were electrophoresed in 7.5%, 10%, or 12.5% SDS-PAGE gels and then transferred to the PVDF membrane (Millipore). After blocking with 5% non-fat milk for 1 h, membranes were incubated with the following primary and secondary antibodies: Flag (1:5000, AP0007, Bioworld, Irving, TX, USA), HDAC3 (1:1000, 3944, Cell Signaling Technology, Danvers, MA, USA), TMEM 173/STING (1:1000, ab92605, Abcam), GADPH (1:1000, AP0063, Bioworld), IRF3 (1:1000, BM4616, Boster, Pleasanton, CA, USA), TBK1 (1:1000, BM4038, Boster), Phospho-TBK1 (1:1000, 5483, Cell Signaling Technology), cGAS (1:1000, 15102, Cell Signaling Technology), PCNA (1:1000, 13110, Cell Signaling Technology), Ki67(1:1000, ab16667, Abcam), TriMethyl-Histone H3-K4 pAb (1:1000, A2357, ABclonal, Woburn, MA, USA), Histone H3 Polyclonal Antibody (1:1000, A2348, ABclonal), Acetyl-Histone H4-K5 pAb (1:1000, A15233, ABclonal), Acetyl-Histone H3-K9 pAb (1:1000, A7255, ABclonal), Acetyl-Histone H3-K27 pAb (1:1000, A7253, ABclonal), DiMethyl-Histone H3-K4 pAb (1:1000, A2356, ABclonal), Acetyl-Histone H3-K4 pAb (1:1000, A17019, ABclonal), Goat Anti-Rabbit IgG-HRP (1:5000, ab6721, Abcam), and Rabbit Anti-Mouse IgG-HRP (1:5000, ab6728, Abcam). Chromogenic detection was performed using the enhanced chemiluminescence system ECL (Millipore) and detected by autoradiography. The Western blot results were quantified using Image J software.

**Co-immunoprecipitation.** The preparation of lysates from a set of cells is as described above. The cell lysates were incubated with protein A/G agarose beads (Cat. #: sc-2003, Santa Cruz) to clean the background for 2 h at 4 °C. After centrifuging at 14,800 rpm for 15 min, the supernatant was collected and incubated with an antibody against STING overnight. After 2-h incubation with protein A/G agarose beads, the precipitates were washed with IP buffer (20 mM HEPES, pH 8.0, 0.2 mM EDTA, 5% glycerol, 150 mM NaCl, and 1% NP-40) 3 times (8 min each). The precipitated proteins were then analyzed with Western blot using indicated antibodies.

**Cell proliferation assay.** ECC1 and Ishikawa cells were seeded into 96-well plates at 5000 cells and maintained overnight before treatment. Cells were assessed using Cell Titer 96 AQ one Solution (Promega) at 0, 24, 48, and 72 h according to the manufacturer’s instruction. In total 20 μL of Cell Titer 96 AQ one Solution was added into each well and the plate was incubated at 37 °C for 1 h. Absorbance was measured at 492 nm.

**Cell migration assay.** After 24 h transient transfection with cGAMP, ECC1 and Ishikawa cells were seeded into FluoroBlok 24-Multiwell insert system (Corning) with a serum-free upper medium. The medium containing 10% FBS was added in the bottom well for 16 h. The upper cells were wiped by swabs and the migrated cells were stained with calcein (Life Technologies, Carlsbad, CA, USA). Ten random fields were captured using a fluorescence microscope and counted using Image J software.

**STING promoter–reporter assay.** For luciferase assay, HEK 293T cells were plated into a 24-well plate, then transfected with a reporter plasmid using Lipofectamine 2000 transfection reagent, a Renilla luciferase plasmid was added as internal reference. After 24 h, RGFP-966 was added to the cells for additional 24 h, and the cells were lysed, and then the luciferase activities were measured using a Dual-Luciferase Reporter Assay System (Promega, Madison, WI, USA).

**Stable cell line construction.** The stability of the STING overexpressed cell lines were screened for virus infection. We first constructed the STING overexpressed lentiviral vector system, transfected into ECC1 and Ishikawa cells together with the vector components for packaging. After the virus suspension was transfected into the cells for 24 h, the common resistance marker of eukaryotic expression vectors, puromycin, was added into the cells for the screening concentration, then we picked out the monoclonal for proliferation and selected for the stability test, and the stable strains of puromycin-resistant monoclonal cells were screened and identified by WB.

**CRISPR/Cas9-mediated knockout of STING.** sgRNA directed against STING were designed at http://crispr.mit.edu and then cloned into the pspCas9-2A-GFP (PX458) vector (Addgene, Watertown, MA, USA) digested with the Bpil Restriction enzyme (NEB). To acquire the knockout cell line, 6 μg of sgRNA-expressing plasmids were transfected into ECC1 or Ishikawa cells using Lipofectamine 2000. After 48 h, GFP-positive cells were sorted by flow cytometry, and then single-cell clones were obtained by limiting dilution. After clonal expansion, STING^−/−^ clones were confirmed by Western blot.

**Colony formation assay.** After being treated with RGFP-966 or transfected with pcDNA3.1-STING, 200 cells were plated into a 3 cm dish in complete medium for 7–10 days. The medium was discarded and cells were washed twice with PBS when the colonies were visible. After being fixed with 95% ethanol for 10 min, Crystal Violet solution was used to stain the colonies for 10 min. Images were taken of the stained dishes, and the number of colonies was counted manually.

**Quantitative real-time PCR.** Total RNA was extracted from cells or human endometrial carcinoma tissues using TRIzol reagent (Invitrogen, Waltham, MA, USA) according to the manufacturer’s instruction. The cDNA was generated with Hiscript II Q RT Supermix for the qPCR kit (Vazyme, Nanjing, China). The qPCR reaction was performed in 20 µL SYBR Green PCR Master Mix (Vazyme, China). GADPH RNA was used for normalization. All the PCR primer sequences are listed in Appendix A.

**Chromatin immunoprecipitation (ChIP) assay.** ChIP assay was performed using a ChIP kit from EpiGentek (Farmingdale, NY, USA) following the manufacturer’s protocol. The immune-precipitations were performed with 2 μg antibodies for HDAC3, H3K4ac, or IgG control, respectively. After de-crosslinking, the antibody-associated DNA was PCR-amplified using the following primers set for human STING promoter: 1F, 5′-GTGCAGTGGCCTAATCTCT-3′ and 1R, GGTGGTGCGGACCGATTAA-3; 2F, TGTTTTTAGTAGAGATGGG-3′ and 2R; 3F, 5′-CTGCAATTACTTTTGCTC-3′ and 3R, 5′-GAGGGGCACAGAGAGGAAT-3′. PCR products were analyzed by qPCR.

**Cell apoptosis assay.** Flow cytometric cell apoptosis analysis was performed using Alexa Fluor 488 Annexin V/Dead Apoptosis kit (Thermo Fisher) according to the manufacturer’s instruction. Briefly, cells were divided into 12-well plates and treated with cGAMP (Sigma, St. Louis, MO, USA) or RGFP-966 (MedChemExpress, Monmouth Junction, NJ, USA) for the indicated time. Cells were harvested and resuspended with binding buffer, following staining with Alexa Fluor 488 annexin V and PI for 15 min at 37 °C. Cell suspensions were analyzed by cytometry (BD, Franklin Lakes, NJ, USA). For each test, at least 10,000 events were counted. Results were analyzed using FlowJo software.

**Mouse xenograft assays**. The protocol conformed to the Guidelines for the Care and Use of Laboratory Animals of Tongji University, and approved by the Animal Care and Ethics Review Committee. The experimental group was determined by randomization. Twelve female BALB/c nu/nu mice (4–6 weeks old) were randomly selected and divided into two groups. 1 × 10^6^ ISK cells or 1 × 10^6^ STING overexpression stable ISK cells were suspended in 100 μL PBS and then injected into the mice. Two weeks later, the tumor was checked for presence. ISK cells and STING overexpression stable ISK cells were injected and analyzed for their abilities to form xenograft tumors.

**Statistical analysis.** All quantitative data are expressed as mean ± standard deviation (SD). The significance of the differences in mean values is analyzed by the Unpaired Student’s *t*-test between two groups and ANOVA for more than two groups. *p* < 0.05 and *p* < 0.01 were considered significant.

## 3. Results

### 3.1. The STING Expression Is Suppressed in Endometrial Carcinoma Tissues

STING expression is suppressed in gastric and colorectal carcinoma cells but up-regulated in tongue squamous cells [14,20]. To determine whether the expression of STING varies with EC progression, we first analyzed the STING expression in EC tissues collected from patients. STING expression decreased progressively with the increasing severity of pathology in immunocytochemistry (IHC) stained sections (Figure 1A). In EC patients, the protein (Figure 1B,C) and mRNA (Figure 1D) levels of STING decreased more than adjacent normal endometrial tissue. To further verify this decrease, we analyzed large EC samples from TCGA databases. The results (Figure 1E,F) are consistent with our results: the STING expression decreased in EC, and the decrease was related to cancer grade. Furthermore, the analysis revealed that the STING expression level is closely related to tumor recurrence; non-recurrence patients had higher STING expression (Figure 1G). Collectively, the STING expression is suppressed in EC patients.

### 3.2. Activated STING Inhibits Proliferation and Facilitates Apoptosis in Endometrial Carcinoma

STING signaling has become a potent antitumor therapy in many immune-related cancers because of its well-established role in innate immune responses [21]; however, the function of STING in tumor cells is unclear. We next investigated the potential role of STING in tumor traits associated with tumor progression. To effectively activate STING signaling, we first added a STING agonist 2′3′-cGAMP to ECC1 and Ishikawa cells. As anticipated, 2′3′-cGAMP significantly activated the STING pathway, as detected by increased phosphorylation of TBK1 and IRF3, two typical STING downstream factors (Figure 2A). Activated STING signaling inhibited cell proliferation as detected through CCK8 assays in ECC1 and Ishikawa cells (Figure 2B,C). Moreover, 2′3′-cGAMP-activated STING signaling significantly promoted the apoptosis of ECC1 and Ishikawa cells (Figure 2D,E). However, transwell assays revealed that activated STING signaling had no significant differences in cell migration (Figure 2F,G). We then detected the activated STING signaling pathway in tumorigenesis in EC cells and identified the STING effect on tumorigenicity through a mouse xenograft assay (Figure 2H). Ishikawa (control) cells and cells with STING overexpression stable cell lines were injected into BALB/c nu/nu mice. The tumors were relieved and visible in the STING-activated group (Figure 2I). However, the effect of STING activation in EC is unclear. To determine whether the expression of STING has the same function as 2′3′-cGAMP in EC cells, we constructed a Flag-tagged pcDNA3.1-STING plasmid and transfected it into ECC1 or Ishikawa cells to overexpress STING (Figure 3A). Consistently, overexpressed STING effectively inhibited the proliferation of ECC1 (Figure 3B) and Ishikawa cells (Figure 3C) at 24, 48, and 72 h. The colony formation assay also confirmed that elevated STING expression inhibited the colony formation of ECC1 (Appendix A) and Ishikawa cells (Appendix A). Altogether, these data strongly suggest the role of STING activation and overexpression in EC cancer cellular processes, such as antiproliferation and pro-apoptosis.

### 3.3. HDAC3-Selective Expression or Inhibition Governs the Cell Proliferation and Apoptosis of Endometrial Carcinoma

To further determine the functional significance of HDAC3 in the regulation of EC, we detected that HDAC3 mRNA is aberrant and highly expressed in EC patients (Figure 3D). We then detected the activated HDAC3 signaling pathway in EC cells undergoing tumorigenesis. We constructed a Flag-tagged pcDNA3.1-HDAC3 plasmid and transfected it into ECC1 or Ishikawa cells to induce HDAC3 overexpression (Figure 4B). Consistently, overexpressed STING effectively inhibited the proliferation of ECC1 (Figure 3B) and Ishikawa cells (Figure 3C) at 24 h, 48 h, and 72 h. To evaluate the therapeutic efficacy of HDAC3 inhibition on endometrial cancer, ECC1 and Ishikawa cells were treated with the HDAC3-selective inhibitor RGFP-966. Consistent with the results of activation or overexpression of STING, RGFP-966 inhibited the proliferation of ECC1cells at 24 h, 48 h, and 72 h (Figure 3E), as well as colony formation (Figure 3F). Western blot analysis of tumor cell proliferation markers Ki67 and PCNA (Figure 3G) further confirmed that RGFP-966 effectively inhibited the proliferation of EC cells. Furthermore, the number of apoptotic cells was significantly increased after HDAC3 inhibition (Figure 3H,I). These results also suggest that HDAC3 inhibition may have antiproliferation and pro-apoptosis effects on EC cells.

### 3.4. Both β-Estradiol and HDAC3 Inhibit STING Expression

ERα is a ligand-activated transcription factor that responds to estrogen and acts as a majority oncogenic signaling in EC [22]. We strikingly discovered that STING protein was significantly decreased by β-estradiol stimulation in HEC1A (Appendix A), ECC1, and Ishikawa (Figure 4A) cells. Given the antitumor effects of HDAC3 in EC, epigenetic histone methyltransferase HDACs have been reported to be a corepressor of ERα to regulate gene expression [23]. We explored the possible relationship of acetylation alteration in STING reduction in EC through the overexpression of HDAC3 and then by detecting the reduced STING expression (Figure 4B) in ECC1 and Ishikawa cells. In contrast, reduced HDAC3 using shRNAs significantly increased STING expression (Figure 4C). We then treated EC cells with different doses of HDAC inhibitors, trichostatin (TSA) (Figure 4D), and vorinostat (SAHA) (Figure 4E). TSA and SAHA are effective inhibitors of Class 1 and II HDAC families [24], and SAHA was approved by the FDA for the treatment of CTCL in 2006. The results indicated that TSA and SAHA markedly increased the expression of the STING protein, suggesting that the expression of STING is regulated by acetylation. Moreover, HDAC3 inhibitor RGFP-966, especially, up-regulated the expression of STING at both protein (Figure 4F–H) and mRNA levels (Figure 4I). Altogether, these data suggest that STING expression is regulated by β-estrogen-ERα and HDAC3, whereas HDAC3 inhibitor can restore STING expression in EC.

### 3.5. β-Estradiol-ERα Recruits HDAC3 to STING Promoter and Deacetylates H3K4

It is known that HDAC3 functions exclusively by forming complexes with transcription factors/repressors to direct complexes to specific gene promoters [25]. HDAC3 inhibitor RGFP-966 increased the transcription of a human STING promoter–reporter (Figure 5A), demonstrating that HDAC3 regulates STING expression at the transcriptional level. To gain further insights into the functional relevance of the HDAC3 and the β-estradiol-ERα signal, we analyzed the STING promoter sequence using promo. We discovered a probable ERα binding site TGACC in −221/−225 bp (Figure 5B). Furthermore, we performed the Co-IP assay and found that β-estradiol significantly increased the interaction of HDAC3 and ERα in ECC1 cells (Figure 5C). This evidence suggests that β-estradiol promotes ERα and HDAC3 localized in STING promoters. HDAC3 could deacetylate histones at various sites and repress gene expression [26,27,28]. With the successive decreases in the H3K4Ac, the overexpression of HDAC3 did not affect the levels of H4K5Ac, H3K9Ac, H3K27Ac, H3K4me2, and H3K4Ac in EC cells (Figure 5D). An analysis of ChIP-seq data from the GEO database demonstrated that the STING promoter contains an H3K4Ac binding sequence near the transcription initiation site. This information was used to determine whether the H3K4Ac effect conferred the STING repression; ChIP analysis verified that HDAC3 and H3K4Ac were more bound to −307/−433 than −1703/−1845 in ECC1 and Ishikawa cells (Figure 5E). We then constructed a truncated STING promoter–luciferase reporter containing the sites (−400/+229) related to the analyzed H3K4Ac. Luciferase analysis revealed that the HDAC3 inhibitor RGFP-966 significantly induced the transcriptions of both the truncated and full-length plasmids (Figure 5F). These findings suggest that HDAC3 decreases the H3K4Ac level that binds to the STING promoter, thereby reducing the expression of STING, which may be facilitated through β-estradiol-ERα in EC.

### 3.6. STING Is Critical for the Antiproliferation and Pro-Apoptotic Function of HDAC3 Inhibition

To further confirm the critical role of STING restoration induced by the HDAC3 inhibitor RGFP-966 in proliferation and apoptosis, we knocked out *STING* in ECC1 and Ishikawa cells with CRISPR-Cas9 (Appendix A). We further treated WT and STING knockout cells with HDAC3 inhibitor RGFP-966. RGFP-966 significantly inhibited cell proliferation (Figure 6A) and colony formation (Figure 6B) in WT cells; however, an insignificant difference was found after the inhibition of HDAC3 in *STING* knockout cells. Furthermore, the pro-apoptosis effect of RGFP-966 was also restrained when STING was absent (Figure 6C,D). STING knockdown tumor cells exerted a high proliferation marker of Ki67 (Figure 6E). The increased cytokine expression induced by RGFP-966 was largely compromised with CRISPR/Cas9-mediated STING knockdown cells (Figure 6F). These findings indicate that β-estradiol-ERα recruited HDAC3 and induced deacetylation of histone 3 lysine 4 by binding to the STING promoter, thereby reducing the expression of STING, HDAC3 inhibition, and that the subsequent STING activation is a promising therapy for ERα-driven EC (Figure 7).

## 4. Discussion

EC is an epithelial malignant tumor of the female endometrium and a prevalent malignancy of the female reproductive system. The incidence of the disease is increasing annually, and the mortality rate of patients is also increasing, meaning it has become a major disease threatening women’s health. Surgery is the main treatment method for EC, supplemented by radiation therapy, chemotherapy, hormone, and biological therapy. However, these traditional treatment methods cannot fundamentally solve the problems of EC recurrence, metastasis, and treatment tolerance, and they are accompanied by severe side effects. Here, we demonstrated that the expression of STING in endometrial cancer lesions was significantly lower than in control patients (patients without endometrial cancer). We then detected the overexpression and inhibition of STING in EC cell lines, ECC1 and Ishikawa cells, which are the typical endometrial cancer cell lines used to study tumorigenesis, the cell proliferation, and apoptosis in EC, and to further determine the functional relevance of STING in the regulation of EC. Activation of the STING signaling pathway inhibits the proliferation and promotes the apoptosis of endometrial cancer cells. Furthermore, EC is clinically characterized by high estrogen-induced production. Furthermore, we found that the expression of STING is regulated by β-estrogen and acetylation; HDAC3 regulates H3K4 deacetylation through binding to the STING promoter. This is the first study to demonstrate that the STING pathway is inhibited in endometrial cancer, laying groundwork for treating endometrial cancer by combining STING agonists and HDAC3 inhibitors.

cGAS is activated by recognizing double-strand DNA, a prominent antiinflammatory response in bacterial and viral infections. cGAS may recognize apoptotic and necrotic DNA released by mitochondria or damaged cells. Typically, there are two major downstream pathways that cause the activation of the intracellular cGAS–STING pathway. One pathway, the STING–TBK-1–IRF3 axle, and the other, the STING–IKK–NF-κB axle, induce type I IFN responses. Studies have indicated that the expression of STING is frequently lower in some melanoma cell lines and tissues of colorectal carcinoma but higher in tongue squamous cell carcinoma [20]. This study found that the expression of STING and TBK-1 was positively correlated with the expression of cGAMP. P-TBK-1 and P-IRF3 were positively correlated with STING, indicating that the STING–TBK-1–IRF3 signal is the dominant pathway for generating type I IFN responses according to our model. To explore the association of clinical outcomes to STING expression, we evaluated the STING–TBK-1–IRF3 signaling key pathway factors such as cell proliferation and immunohistochemical scores.

Typically, EC is considered to be a sex steroid hormone aberration disease. Uterine dysfunction may be due to local increased high estrogen receptor (ER) expression promoting cancer formation. Indeed, suppressive hormone treatments, such as continuous use of oral contraceptive, high-dose progesterone, levonorgestrel-releasing intrauterine system, danazol, and gonadotropin-releasing hormone agonists, provide beneficial clinical outcomes for EC. However, new reports support the hypothesis that immune-inflammatory responses are also involved in the development of EC. In patients with adenomyosis, the expression of several inflammatory cytokines, such as IL-1β, IL-6, IL-8, IL-10, TNF, NF-κB, MCP-1, and RANTES, is abnormal, and a variety of signaling pathways such as TLR437 and MAPK/ERK are involved. In addition, immune cells, such as macrophages, natural killer cells, and T helper cells, have also been reported to participate in the development of EC. We demonstrated that estrogen repressed STING expression through the acetylation of STING DNA; HDAC3 is the key enzyme in estrogen-reduced STING expression, which is consistent with a previous study showing a correlation between NF-κB DNA-binding activity and dysmenorrhea severity in adenomyosis.

In the progression of EC, epigenetic modifications play an essential role in regulating transcription, DNA repair, and replication [29,30]. Histone modifications are thought to be involved in tumorigenesis, occurring in the early stages of cancer. They regulate many cellular functions, such as cell proliferation, apoptosis, and differentiation. Histone acetylation occurs through the addition of acetyl groups to the lysine residues in histone tails by histone acetyltransferases (HATs, also known as “Writers”) and is reversibly regulated by histone deacetylases (HDACs, also known as “Erasers”) to remove acetyl groups. In EC, histone acetylases (HATs) and histone deacetylases (HDACs) are essential in the regulation of endometrial remodeling [31], suggesting that histone acetyltransferase MOF acts as a potential tumor suppressor and regulates ERα function in EC [32]. Given that HDACs’ deregulation is involved in tumorigenesis, HDAC inhibitors are effective therapies for many cancers. The expression levels of HDACs are higher than that of normal endometrium in EC. HDAC1, 2, and 3 are overexpressed compared with the normal endometrium, and the higher expression is associated with a poor prognosis [33,34]. Other HDACs, the class III HDACs, such as SIRT1, 2, 4, 5, and 6, have lower expression levels in EC, except for SIRT7 [35]. A recent study demonstrated that high SIRT1 expression in patients was associated with better overall survival, suggesting that SIRT1 may act as a tumor suppressor [36]. Herein, HDAC3 inhibitors were found to significantly up-regulate the expression of STING, revealing that HDAC3 is a key acetylated regulatory enzyme that regulates STING expression in endometrial cancer. Therefore, it is essential to improve the targeting effect of HDAC3.

## 5. Conclusions

This study shed light on the cGAS–STING pathway; that its downstream key factors were inhibited in EC patients and that this inhibition is correlated with HDAC3 deacetylation epigenetic regulation of histones, suggesting that the inhibition of HDAC3 can restore the STING protein and ameliorate EC. Combining HDAC3 inhibitors and STING activators may contribute to combating EC pathogenesis.

## Figures and Tables

**Figure 1 cancers-14-04718-f001:**
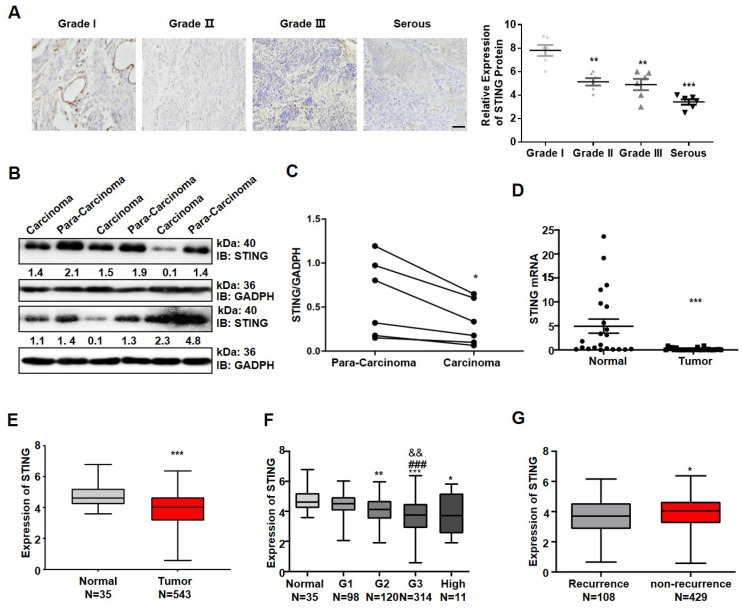
The STING expression is suppressed in endometrial carcinoma tissues. (**A**) Representative immunohistochemical images and the quantification of STING expression levels during various stages of endometrial cancer patients. Original magnification 200×. Scar Bar = 50 μm. (**B**) IB analysis of STING expression levels in various EC tissues lysis. GAPDH was used as a loading control. (**C**) Western blot quantifications of STING protein expression in Figure 1B. (**D**) RT-qPCR analysis of STING RNA in EC tissues (n = 22) and normal tissues (n = 27). (**E**) Relative expression of STING in EC tissues (n = 543) and normal tissues (n = 35) in TCGA databases. (**F**) Analysis of the STING expression during various stages of EC patients in TCGA databases. (**G**) The expression of STING in recurrent (n = 108) and non-recurrent (n = 429) EC in the TCGA database. Data are shown as mean ± SD. Two tail t-tests were used to calculate the *p*-value. ** p* < 0.05, *** p* < 0.001, **** p* < 0.0005 significantly different from control, *^###^ p* < 0.001 significantly different from G1, and ^&&^
*p* < 0.001 significantly different from G3. Original image of Western blot can be found in Appendix A.

**Figure 2 cancers-14-04718-f002:**
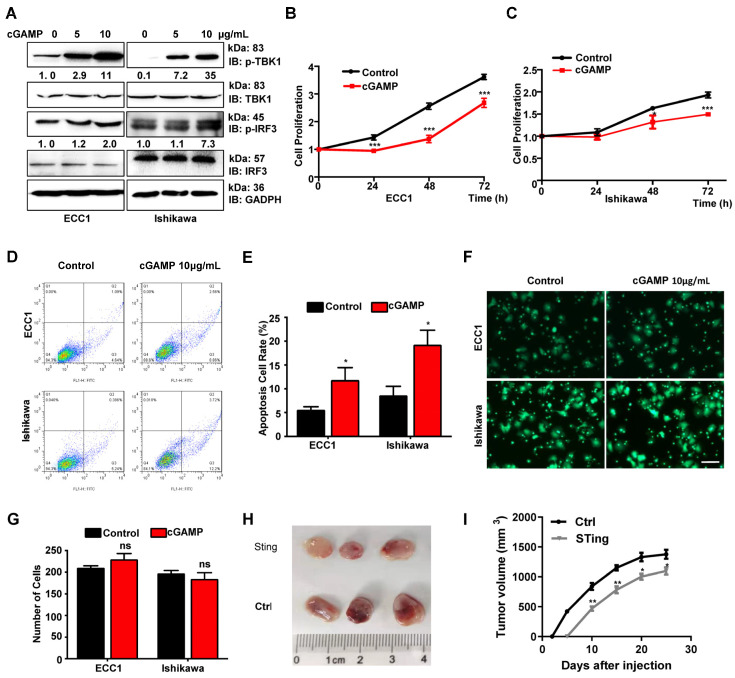
Activated STING inhibits proliferation and facilitates apoptosis in endometrial carcinoma. (**A**) Western blot analysis of the whole cell lysis derived from ECC1 and Ishikawa cells supplemented with 0, 5, or 10 μg/mL cGAMP in the medium for 4 h. (**B**,**C**) CCK-8 analysis of the cell proliferation of ECC1 (**B**) and Ishikawa (**C**) cells transfected with 10 μg/mL cGAMP. (**D**) Flow cytometry analysis of the cell apoptosis of ECC1 and Ishikawa cells transfected with or without 10 μg/mL cGAMP for 48 h. (**E**) Quantification of apoptosis cells in Figure 2D. (**F**) Representative images of transwell analysis of ECC1 and Ishikawa cells transfected with or without cGAMP (10 μg/mL). (**G**) Quantification of migrated cells in Figure 2F. (**H**,**I**) Tumorigenicity of ISK cells and STING overexpression cells detected by morphology (**H**) and xenografts volumes (**I**). The cell-based experiments were repeated at least thrice, and representative results were shown. Data are shown as mean ± SD. Two tails *t*-tests were used to calculate the *p*-value. ** p* < 0.05, *** p* < 0.001, **** p* < 0.0005 significantly different from Control. Original image of Western blot can be found in Appendix A.

**Figure 3 cancers-14-04718-f003:**
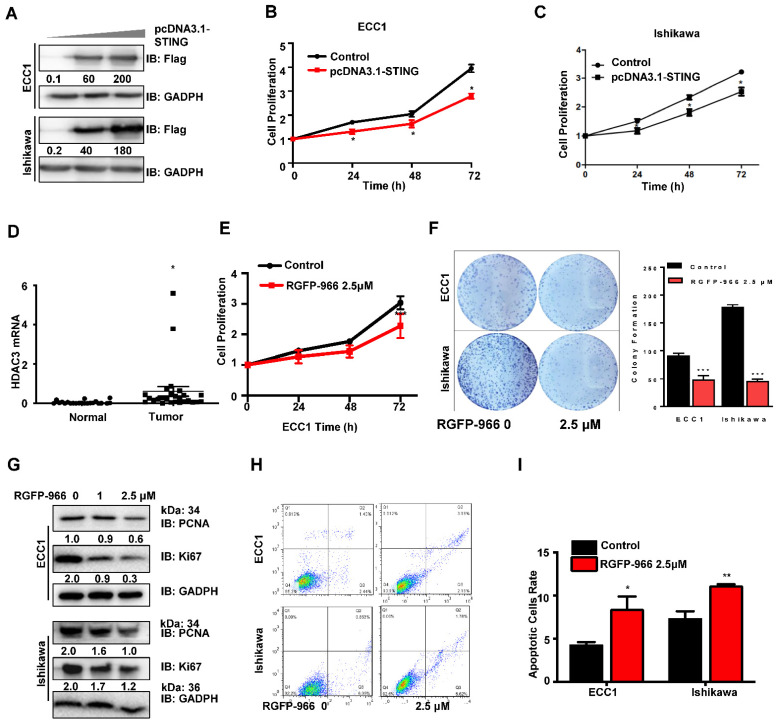
HDAC3-selective expression or inhibition governs the cell proliferation and apoptosis of endometrial carcinoma. (**A**) IB analysis of WCL derived from ECC1 and Ishikawa cells transfected with pcDNA3.1-STING constructs for 24 h. (**B**,**C**) CCK-8 analysis was used to detect the proliferation of ECC1 (**B**) and Ishikawa (**C**) cells transfected with pcDNA3.1-STING construct. (**D**) qPCR analysis of HDAC3 mRNA expression in EC tissues (n = 22) compared with normal tissues (n = 27). (**E**) CCK8 analysis of ECC1 cells treated with 2.5 μM RGFP-966 for 0–72 h. (**F**) Colony formation assay of ECC1 and Ishikawa cells treated with 2.5 μM RGFP-966. (**G**) IB analysis of WCL derived from ECC1 and Ishikawa cells treated with indicated doses of RGFP-966 for 48 h. (**H**) Flow cytometry analysis of ECC1 and Ishikawa cells were transfected with 2.5 μM RGFP-966 for 48 h. (**I**) Quantification of apoptosis cells in Figure 3H. The cell-based experiments were repeated at least three times, and representative results are shown as mean ± SD. Two-way ANOVA analyses were used to calculate the *p*-value. ** p* < 0.05, *** p* < 0.001, **** p* < 0.0005 significantly different from control. Original image of Western blot can be found in Appendix A.

**Figure 4 cancers-14-04718-f004:**
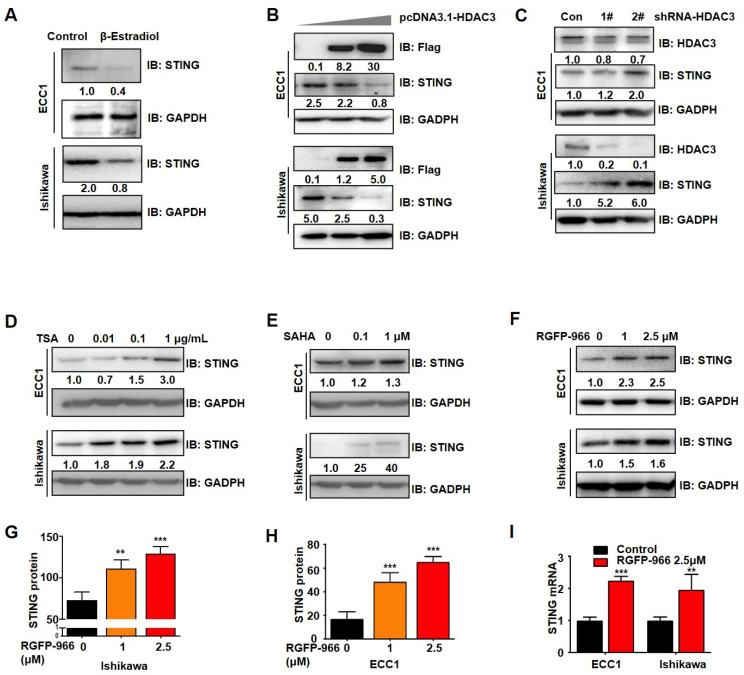
Both β-estradiol and HDAC3 inhibit STING expression. (**A**) IB analysis of WCL derived from HEC1A cells treated with 0–100 nM β-estradiol for 24 h. (**A**) IB analysis of WCL derived from ECC1 cells and Ishikawa cells treated with 10 nM β-estradiol for 24 h. (**B**,**C**) IB analysis of WCL derived from ECC1 and Ishikawa cells. Cells were transfected with pcDNA3.1-HDAC3 (**B**) constructs or shRNA-HDAC3 (**C**) for 24 h. (**D**,**E**) ECC1 and Ishikawa cells were treated with TSA (0, 0.01, 0.1, and 1 μg/mL) for 24 h (**D**), with increasing doses of SAHA (0, 0.1, and 1 μM) for 24 h (**E**), and with increasing doses of RGFP-966 for 24 h (**F**). (**G**,**H**) Protein quantification of STING using Image J software. (**I**) qPCR analysis of ECC1 and Ishikawa cells treated with 2.5 μM RGFP-966 for 24 h. The cell-based experiments were repeated at least three times, and representative results are shown as mean ± SD. *t*-test analyses were used to calculate the *p*-value. *** p* < 0.001, **** p* < 0.0005 significantly different from control. Original image of Western blot can be found in Appendix A.

**Figure 5 cancers-14-04718-f005:**
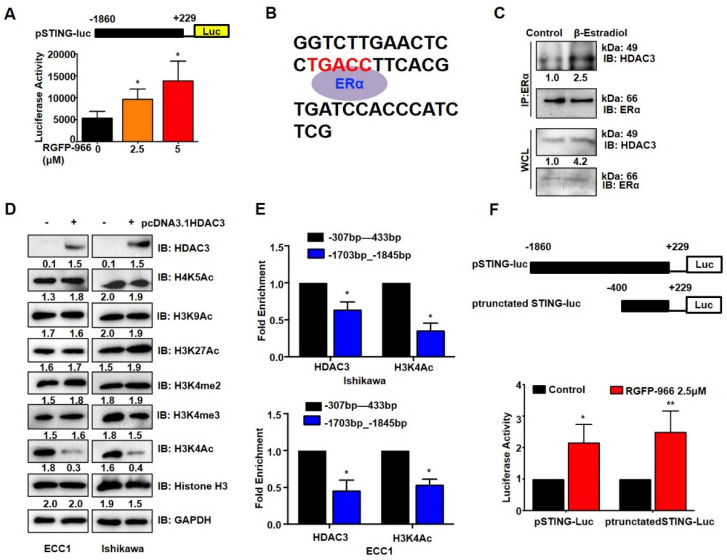
β-estradiol-ERα recruits HDAC3 to STING promoter and deacetylates H3K4. (**A**) HEK 293T cells were transfected with STING promoter–reporter plasmid for 24 h, the *Renilla* plasmid as control; cells were treated with an increasing dose of RGFP-966 for another 24 h, and luciferase activities were measured. (**B**) The DNA sequence of the STING promoter was analyzed using promo. (**C**) IB analysis of WCL and IP derived from Ishikawa cells treated with 10 nM β-estradiol for 24 h. (**D**) IB analysis of WCL derived from Ishikawa and ECC1 cells transfected with pcDNA3.1-HDAC3 for 24 h. (**E**) Ishikawa (up) and ECC1 (down) cells were subjected to ChIP assay using HDAC3 or H3K4Ac antibodies. Immunoprecipitated DNA was measured by qPCR using primer for human STING promoter (−307_−433, −1703_−1845). (**F**) HEK 293T cells were transfected with control STING promoter–reporter and truncated STING promoter–reporter plasmid, respectively, plus a *Renilla* plasmid as an internal reference for 24 h, then treated with 2.5 μM of RGFP-966 for 24 h, and then subjected to luciferase activity assay. Original image of Western blot can be found in Appendix A. * *p* < 0.05, ** *p* < 0.001.

**Figure 6 cancers-14-04718-f006:**
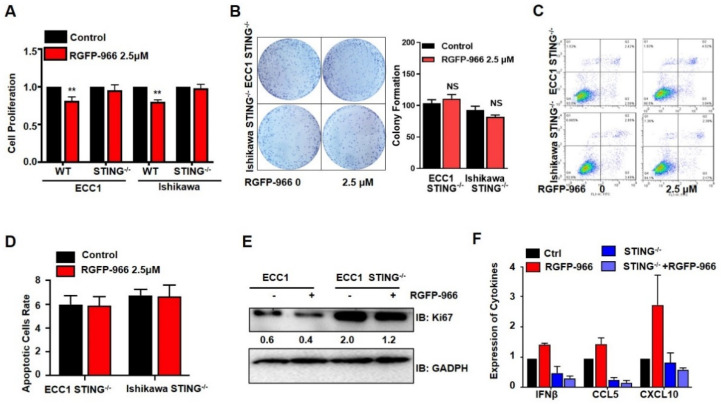
STING is critical for the antiproliferation and pro-apoptotic function of HDAC3 inhibition. (**A**) CCK8 analysis of ECC1 cells WT or STING^−/−^ and Ishikawa cells WT or STING^−/−^ treated with 2.5 μM RGFP-966 for 48 h. (**B**) Colony formation analysis of ECC1 and Ishikawa cells transfected with 2.5 μM RGFP-966. (**C**) Flow cytometry analysis of ECC1 cells WT or STING^−/−^ and Ishikawa cells WT or STING^−/−^ treated with 2.5 μM RGFP-966 for 48 h. (**D**) Apoptotic analysis of ECC1 cells WT or STING^−/−^ and Ishikawa cells WT or STING^−/−^ treated with 2.5 μM RGFP-966 for 48 h. (**E**) IB analysis of WCL derived from ECC1 cells WT or STING^−/−^ and Ishikawa cells WT or STING^−/−^ treated with 2.5 μM RGFP-966 for 48 h. (**F**) qPCR analysis of cytokines from ECC1 cells WT or STING^−/−^ and Ishikawa cells WT or STING^−/−^ treated with 2.5 μM RGFP-966 for 48 h. The cell-based experiments were repeated at least three times and representative results are shown as mean ± SD. Two-way ANOVA analyses were used to calculate the *p*-value. *** p* < 0.001, significantly different from control. Original image of Western blot can be found in Appendix A.

**Figure 7 cancers-14-04718-f007:**
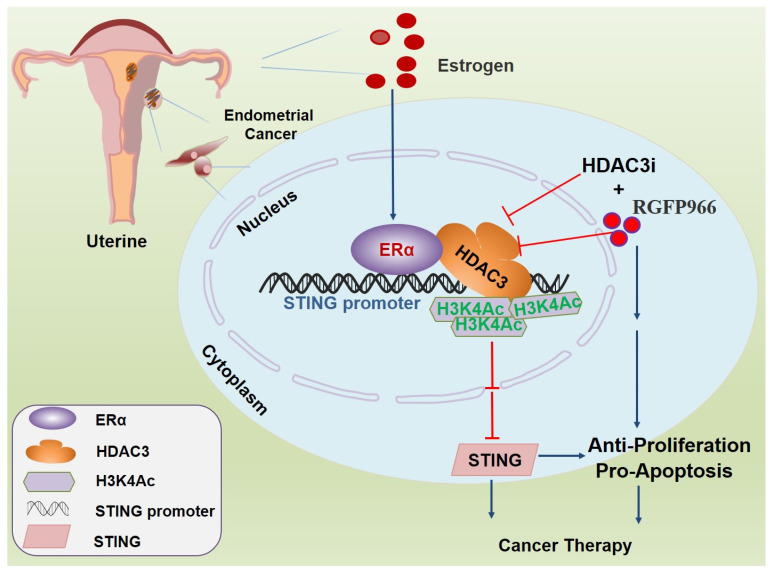
Mechanism of HDAC3-dependent expression of STING in β-estradiol-ERα-mediated endometrial tumorigenesis. In endometrial cancer, the stimulation of interferon genes STING decreased in EC, which is a key regulator of cell proliferation, apoptosis, and fate. Activating the STING signaling pathway can decrease proliferation and induce apoptosis in EC cells. Moreover, HDAC3-selective inhibition preferentially inhibits proliferation, facilitates apoptosis in EC, and derepresses STING expression, whereas β-estradiol-ERα, one of the highest risk factors of EC, might facilitate the repression of STING. In endometrial cancer cells, knockdown or inhibition of the HDAC3 by an HDAC3-selective inhibitor RGFP966 increases STING expression, thereby inhibiting tumorigenesis. Mechanically, we identified that β-estradiol-ERα recruited HDAC3 and induced the deacetylation of histone 3 lysine 4 by binding to the STING promoter, thereby reducing the expression of STING. These findings suggest a novel molecular mechanism underlying HDAC3, and the subsequent STING suppression constitutes an important regulatory loop that promotes ERα-driven endometrial tumorigenesis, and the use of HDAC3-selective inhibitors may be an effective treatment for endometrial cancer.

## Data Availability

The data generated or analyzed during this study are included in this paper and its Appendix A. The data supporting the findings in the main text are found in Appendix A.

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
