# Peer review of "Histone Deacetylase 3 Governs β-Estradiol-ERα-Involved Endometrial Tumorigenesis via Inhibition of STING Transcription"

_cancers, 2022, doi:10.3390/cancers14194718_

Round 1
Reviewer 1 Report
In the article "Histone deacetylase 3 inhibits STING transcription and gov- 1 erns β-Estradiol-ERα mediated endometrial tumorigenesis", authors demonstrated a novel molecular mechanism underlying HDAC3 inhibited STING transcription via β-Estradiol-ERα and suggested combined targeting of HDAC 25 and STING as a promising therapy for combating endometrial cancer.
Overall, study looks interesting and promising but representation of data is not sufficient. Its hard to believe if the data has been independently repeated thrice.
For each western blot, densitometry is required.
representation of raw western blot in supplementary file should be improve to draw conclusion and avoid any fabrication.
I strongly recommend to repeat all data thrice before submitting it for publication.
also One more cell line should be added for analysis.
On what basis author choose ECC1 and Ishikawa should be added in the discussion.
here are some minor concern:
Figure 2 Legend: Cells were transfected with 0,5 or 10 ug/ml cGAMP?? is it correct?
for Apoptosis assay: %age of cell is required from triplicate experiment
Author Response
Response: We thank the reviewer for recognizing the significance of our study, we also thank the reviewer for the kindly suggestions to guide us to improve this project. We have measured the quantification and added it to the results. We have quantified all Western blots by a statistic analysis of densitometry and colony formation assays in figures and supplement files. Moreover, we have added the discussion of choose ECC1 and Ishikawa on page 14, line 478, to further determine the functional relevance of STING and HDAC3 in the regulation of EC, we detected the overexpression and inhibition of STING and HDAC3 in EC cell lines, ECC1 and Ishikawa cells, which are the typical endometrial cancer cell lines to study tumorigenesis, the cell proliferation and apoptosis in EC. thank you.
here are some minor concern:
Figure 2 Legend: Cells were transfected with 0,5 or 10 ug/ml cGAMP?? is it correct?
Answer: Thank you for the suggestion. We have reorganized the manuscript and the legends, and rewritten the sentence with “cells supplemented with 0, 5, or 10 mg/mL cGAMP in the medium for 4h”.
for Apoptosis assay: %age of cell is required from triplicate experiment.
Answer: Thank you for the suggestion. The percent of apoptosis assay was from triplicate experiments and we have added the original image of the FACS results in Supplement Files.
Reviewer 2 Report
The manuscript entitled “Histone deacetylase 3 inhibits STING transcription and gov-1 erns β-Estradiol-ERα mediated endometrial tumorigenesis” by Chen et al., examined the role of STING transcription against endometrial tumorigenesis. I found the study very interesting and well written. However, I recommend the manuscript to go through a major revision and address the following concerns:
- Authors need to modify the cell culture section described in the manuscript. I found that Hec1A cell line has been used in the study, however, I couldn’t find it in the description.
- Similarly, authors have mentioned the usage of HEK293T cells, but not described the purpose in the manuscript.
- How did authors confirm the stability of the overexpressed cell line? And I also recommend to mention the methodology used for making the stable cell line.
- I strongly recommend to indicate the molecular weight of each protein band.
- I also wonder whether vector alone (without target gene: SCR) were taken under consideration in the experiment as a negative control or not? For example, in figure 2B-C, 4B, 5D and so on. If not, I highly recommend to conduct the experiment using the suitable controls to understand the results more clearly.
- What is the rationale for the delay in the injection of STING overexpressed cell line to develop the xenograft mice model? As the results depicted, I think there may not be any significant difference in the tumor volume if the cells were injected on the same day.
- I also suggest performing H&E staining in tumor biopsy to confirm the overexpression of STING retains throughout the study time period.
- In figure 4C, negative control is missing.
Author Response
Answer: We thank the reviewer for recognizing the significance of our study, we also thank the reviewer for the kindly suggestions to guide us to improve this project. We have modified the cell culture section described in the manuscript, we have added the Hec1A cell line in the description, and mentioned the purpose of the usage of HEK293T cells in the manuscript on page 3, line 347 “The human endometrial carcinoma cell lines ECC1, Ishikawa and Hec 1A cells in the lab were kindly from the Key Laboratory of Maternal Fetal Medicine, Clinical and Translational Research Center, and were confirmed using STR profiling, and HEK293T cells were used for the virus packaging of shRNA”.
How did authors confirm the stability of the overexpressed cell line? And I also recommend to mention the methodology used for making the stable cell line.
Answer: Thank you for the suggestion. We have added the stability of the overexpressed cell line in the method on page 4, line 193 “The stability of the STING overexpressed cell lines were screened for virus infection, we first constructed the STING overexpressed lentiviral vector system, transfected into ECC1 and Ishikawa cells together with the vector components for packaging, after the virus suspension was transfected into the cells for 24 hours, the common resistance marker of eukaryotic expression vectors, puromycin, was added into the cells for the screening concentration, then picked out the monoclonal for proliferation and selected for the stability test, the stable strains of puromycin resistant monoclonal cells were screened and identified by WB”.
I strongly recommend to indicate the molecular weight of each protein band.
I also wonder whether vector alone (without target gene: SCR) were taken under consideration in the experiment as a negative control or not? For example, in figure 2B-C, 4B, 5D and so on. If not, I highly recommend to conduct the experiment using the suitable controls to understand the results more clearly.
Answer: Thank you for the suggestion. We have added the molecular weight of each protein band of the figures, and the used negative control in the figures is the empty vector alone in the results.
Reviewer 3 Report
The study conducted by Guofang Chen entitled “Histone deacetylase 3 inhibits STING transcription and governs β-Estradiol-ERα mediated endometrial tumorigenesis” reported role STING expression endometrial tumorigenesis they found that STING expression was largely reduced in endometrial carcinoma. Activation or overexpression of STING reduce the cell proliferation and induced apoptosis. Author also found that STING expression was deregulated by β-Estradiol and HDAC3 in endometrial carcinoma (EC). The study has been designed properly and most of the results and their conclusions are very convincing and sound. Still, some minor changes would help the readers grab the story better. As per my consideration, it needs minor revision before final acceptance.
Comments
1. Author reported the role of HDAC3 in regulation of STING expression. It will be interesting to test the role of other HDACs in the STING expression.
2. In Figure 4C knockdown of HDAC3 is not efficient in ECC1 cell line. And the effect on STING expression is also confusing (shRNA1-HDAC3 shows better knockdown efficiency in comparison to shRNA2-HDAC3 but have no effect on STING expression and in case of shRNA2 knockdown efficiency is poor but we see change in STING expression please explain). I suggest repeating the experiment and replace these images with improved data.
3. The title needs to be tailored in a more meaningful way.
Author Response
Response: We thank the reviewer for recognizing the significance of our study, we also thank the reviewer for the kindly suggestions to guide us to improve this project. We have modified the manuscript carefully.
Comments
- Author reported the role of HDAC3 in regulation of STING expression. It will be interesting to test the role of other HDACs in the STING expression.
Answer: Thank you for the suggestion. We have concerned the role of other HDACs in the STING expression. Though the link between HDAC and STING was not direct, there were still some connections between HDAC and the cGAS pathway. It reported that HDAC3 could transcriptionally regulate the expression of cGAS and promote cGAS transcription by deacetylating p651. HDAC9 directly enhances activation of the kinase activity of TBK12. HDAC4 repressed the translocation of transcription factor IRF3 to the nucleus, thereby decreasing IRF3-mediated IFN-β expression3. And also, the TSA (HDAC1 inhibitor) induces DNA damage response and depends on cGAS activation 4.
- Liao, Y.; Cheng, J.; Kong, X.; Li, S.; Li, X.; Zhang, M.; Zhang, H.; Yang, T.; Dong, Y.; Li, J.; Xu, Y.; Yuan, Z., HDAC3 inhibition ameliorates ischemia/reperfusion-induced brain injury by regulating the microglial cGAS-STING pathway. Theranostics 2020, 10 (21), 9644-9662.
- Li, X.; Zhang, Q.; Ding, Y.; Liu, Y.; Zhao, D.; Zhao, K.; Shen, Q.; Liu, X.; Zhu, X.; Li, N.; Cheng, Z.; Fan, G.; Wang, Q.; Cao, X., Methyltransferase Dnmt3a upregulates HDAC9 to deacetylate the kinase TBK1 for activation of antiviral innate immunity. Nat Immunol 2016, 17 (7), 806-15.
- Yang, Q.; Tang, J.; Pei, R.; Gao, X.; Guo, J.; Xu, C.; Wang, Y.; Wang, Q.; Wu, C.; Zhou, Y.; Hu, X.; Zhao, H.; Wang, Y.; Chen, X.; Chen, J., Host HDAC4 regulates the antiviral response by inhibiting the phosphorylation of IRF3. J Mol Cell Biol 2019, 11 (2), 158-169.
- Yu-K.; Sheng L.; Lei Z.; Wen.; Jia Pa.; Chao Z.; Bo W.; Jiang W.; Yu S.; Guo Y.; Bei C.; Inhibition of histone deacetylase 1 suppresses pseudorabies virus infection through cGAS-STING antiviral innate immunity. Mol Immunol. 2021 Aug; 136:55-64.
- In Figure 4C knockdown of HDAC3 is not efficient in ECC1 cell line. And the effect on STING expression is also confusing (shRNA1-HDAC3 shows better knockdown efficiency in comparison to shRNA2-HDAC3 but have no effect on STING expression and in case of shRNA2 knockdown efficiency is poor but we see change in STING expression please explain). I suggest repeating the experiment and replace these images with improved data.
Answer: Thank you for the suggestion. We have replaced the images with improved data in figure 4C.
- The title needs to be tailored in a more meaningful way.
Answer: Thank you for the suggestion. We have tailored the title with “Histone deacetylase 3 governs β-Estradiol-ERα-involved endometrial tumorigenesis via inhibition of STING transcription”.
Round 2
Reviewer 2 Report
None